# Study of Morphology, Rheology, and Dynamic Properties toward Unveiling the Partial Miscibility in Poly(lactic acid)—Poly(hydroxybutyrate-co-hydroxyvalerate) Blends

**DOI:** 10.3390/polym14245359

**Published:** 2022-12-07

**Authors:** Hu Qiao, Abderrahim Maazouz, Khalid Lamnawar

**Affiliations:** Univ Lyon, CNRS, UMR 5223, Ingénierie des Matériaux Polymères, INSA Lyon, Université Claude Bernard Lyon 1, Université Jean-Monnet, F-69621 Villeurbanne, France

**Keywords:** rheology, miscibility, morphology, biopolymer blends

## Abstract

The purpose of the present work was to gain a fundamental understanding of how the composition and physico-chemical properties affect the rheology, morphology, miscibility, and thermal stability of poly(lactic acid) (PLA)—poly(hydroxybutyrate-co-hydroxyvalerate) (PHBV) biopolymer blends obtained by melt mixing. First, restricted processing conditions were chosen, due to the inherent thermal degradation of PHBV, as proven by rheological dynamic time sweep (DTS) measurements and size-exclusion chromatography (SEC). Based on this, the composition dependence of the blends was investigated using small-amplitude oscillatory shear rheology (SAOS), and the results were confirmed by scanning electron microscopy (SEM) analysis. Subsequently, the changes in glass transition temperatures (T_g_s) from the molten to the solid state, as observed by DMA and DSC, were verified by coupling SAOS to dielectric relaxation spectroscopy (DRS). Herein, the thermo-rheological complexity of PLA/PHBV blends in the melt was revealed, especially for PLA-rich blends. Irregularly structured morphologies, caused by highly mismatched viscoelastic properties, illustrated the degree of partial miscibility. Moreover, the thermo-rheological complexity appeared in the molten state of the asymmetric PLA-rich phases could be correlated to the crystal-amorphous interfacial MWS polarization, because of the locally-induced phase separation and heterogeneity, and owing to the differences in their crystallization properties during cooling. The miscibility also suffered from the lower thermal stability of PLA and the even more unstable PHBV. Nevertheless, the melt-induced degradation process of the PLA/PHBV blends seemed to be responsible for some of the in situ self-compatibilization and plasticization mechanisms. As a result, the miscibility and thermo-rheological simplicity were improved for the intermediate and PHBV-rich compositions at low temperatures, since their properties were, to a large extent, governed by the significant degradation of PHBV. The present findings should increase the understanding of morphological changes in PLA/PHBV blends and help control their micro/nanostructure.

## 1. Introduction

Currently, the environmental pollution caused by petroleum-based plastics is becoming more and more serious. Bio-based polymers are alternatives to their petroleum-based counterparts and represent an interesting and growing market [1,2]. Poly(lactic acid) (PLA) is a bio-degradable polymer with widespread use, primarily because of its good mechanical properties, relatively low price, and transparency, but unfortunately it has relatively high processing characteristics [3,4,5,6,7,8]. To help overcome this issue, it can be blended with different kinds of polymers, such as PBAT [9], PCL [10], and PHBV [11].

Among these, poly(hydroxybutyrate-co-hydroxyvalerate) (PHBV) can be directly produced by bacterial fermentation of sugars or lipids [12]. Both PLA and PHBV are fully bio-based and biodegradable polyesters, making them environmentally-friendly. Moreover, thanks to the high degree of crystallization of PHBV, it is one of the best candidates for improving the barrier properties of PLA, making composites fabricated with PLA and PHBV very interesting for advanced food packaging [13].

The performance of PLA/PHBV blends has been reported in various recent papers. Tatiana Budtova et al. [14] found suitable processing conditions after investigating the thermal instabilities of PHBV by rheological dynamic time sweep measurements. From a rheological point of view, the interfaces between PLA and PHBV played an important role in their rheological properties in the molten state. The blends followed the mixing law at medium and high frequencies, whereas they showed a decrease of viscosity modulus at low frequencies [14,15,16]. Despite these findings, the rheological properties of PLA/PHBV blends in the molten state, as an instructional tool, have yet to be systematically and comprehensively investigated in relation to their physico-chemical properties and changes during the mixing step.

Owing to their minor difference in solubility parameter values, a good miscibility between PLA and PHBV is thermodynamically expected [17,18]. Previous studies reported [19] that this miscibility is dependent on the molecular weight of each of the two biopolymers. The best miscibility should appear during blending of PLA and PHBV with lower molecular weights, as they are partially miscible when the one of them has a low molecular weight. Blends of PLA and PHBV, in which both materials were high molecular weight polymers showed obvious biphasic separation and two experimental glass transition temperatures T_g_s [16], which is usually taken as proof of immiscibility. Furthermore, morphological observations of the blends demonstrated a nodular structure in the minor phase containing less than 30 wt.% of one component, and a co-continuous structure in the intermediate phases of both polymers [14]. Similarly, the degrees of immiscibility or miscibility have not been studied as a function of the component ratios.

Valentina and co-workers [20] found that barrier properties were improved during the transition from PLA to PHB for all component ratios, owing to the higher degree of crystallization of PHB, which showed a fairly simple superposition of the blending aspect. Boufarguine and co-workers [21] reported that nano-structural co-extrusion of PLA and PHBV improved both the gas barrier and mechanical properties compared with classical blending and three-layer co-extrusion processing. However, it is worth noting that PHBV was broken into lamellas when hundreds of layers were prepared with asymmetric ratios (PLA/PHBV = 90/10).

Above all, for PLA/PHBV blends, although several works on various subjects already exist, systematic and deep studies are still lacking when it comes to properties such as rheology, miscibility, and thermal instability, all of which are significant for the application of PLA/PHBV blends.

Despite the interesting papers in the literature, few works are dedicated to the study of composition dependence, temperature, and time effects on the change of PLA and PHBV properties and the morphology induced during melt mixing. This work, thus, presents a fundamental and systematic investigation of the compositional dependence of various properties of PLA/PHBV blends obtained by melt mixing over the entire composition range. First, thermal degradation was studied using rheological dynamic time sweep measurements and size exclusion chromatography (SEC) (i.e., molar masses analysis). Following this, the primary objectives were to understand the compositional dependence of the thermo-rheological complexity and miscibility, using the results and analysis of viscoelastic properties through small-amplitude oscillatory shear rheology (SAOS). Then, the thermal and thermomechanical properties were investigated by differential scanning calorimetry (DSC) and dynamic mechanical analysis (DMA), respectively. The obtained results with regard to morphological structures and the phase inversion process were then confirmed by scanning electron microscopy (SEM). Finally, the interfacial properties were studied using a rheological Palierne model. Dielectric relaxation spectroscopy (DRS) in both a molten and solid state was presented, which further explained the local heterogeneity behavior and chain dynamics.

## 2. Materials

Poly(lactic acid) (Ingeo 2003D) produced by NatureWorks Co., Ltd., Plymouth, MN, USA was a commercial product. PHBV (Enmat Y1000P) produced by Tianan Biological Materials Co. (Ningbo, China) was provided by Natureplast, France. The pelletized grade used in this study contained a certain amount of stabilizers and nucleating agents, but no details were provided. Their specific chemical structures and basic properties of both biopolymers are shown in Figure 1 and Table 1.

## 3. Preparation of Biopolymer Blends

The pure PLA and PHBV were dried in a vacuum oven at 50 °C for 8 h, to remove moisture prior to blending. Both the pure PLA and PHBV, as well as their blends with compositions ranging from 100/0 to 0/100 (wt.%/wt.%), in increments of 10 wt.%, were melted and melt mixed in a twin-screw extruder (15-mL Microcompounder, DSM Xplore) under nitrogen atmosphere. Specific processing conditions with the lowest possible temperature, i.e., 175 °C for 3 min at a rotor speed of 50 rpm, were chosen, in order to minimize the thermal degradation of PHBV [14]. These parameters were determined beforehand by rheological dynamic time sweep measurements, TGA and SEC, as shown in the next thermal stabilities part. Following melt mixing, disk-shaped samples with a thickness of 2 mm and a diameter of 25 mm were fabricated via the extrusion injection method (Micro 5 cc injection molder, DSM at 1.3 MPa). The temperatures of the injection zone and mold holder were set at 180 °C and 60 °C, respectively.

## 4. Characterization Methods

### 4.1. Differential Scanning Calorimetry

The crystallization properties of PLA/PHBV blends were explored using a differential scanning calorimeter (DSC) (Q23) from TA Instruments (New Castle, DE, USA), with nitrogen as the purging gas at a flow rate of 50 mL/min. Samples of approximately 5 mg were heated from −20 to 190 °C at a rate of 10 °C/min, and the temperature was maintained at 190 °C for 3 min, to eliminate the sample’s thermal history. Subsequently, the samples were cooled down to −20 °C at a rate of 10 °C/min, and reheated to 190 °C at a rate of 10 °C/min. The glass transition temperature ($T_{g}$), enthalpies of melting ($\Delta H_{m}$) and enthalpies of cold crystallization ($\Delta H_{cc}$) were recorded from the second heating cycle, and enthalpies of crystallization ($\Delta H_{c}$) were noted from the cooling process.

The degree of crystalline ($X_{c}$) was determined using the following Equation (1):(1)$$X_{c}=\frac{\Delta H_{m}-\Delta H_{cc}}{\emptyset \Delta H_{m}^{0}}\times 100\%$$

Here, $\Delta H_{m}$ is the heat of fusion of component, $\Delta H_{m}^{0}$ corresponds to the heat of fusion for 100% crystalline polymer, ∅ is the % of PLA or PHBV in the blends, and the value of $\Delta H_{m}^{0}$ = 93.6 J/g [22] and 146.0 J/g [23] for 100% crystalline PLA and PHBV, respectively.

### 4.2. Thermogravimetric Analysis (TGA)

Thermogravimetric analysis (TGA) was performed on a TA Instruments Q500 apparatus under nitrogen atmosphere with a sample mass of 5–10 mg at a heating rate of 10 °C/min under a nitrogen flow of 50 mL/min.

### 4.3. Size Exclusion Chromatography (SEC)

The molecular weight and polydispersity of neat PLA, PHBV, and their blends were determined by size exclusion chromatography (SEC) in an apparatus equipped with 3 Waters HR5E columns, using hexafluoroisopropanol (HFIP) as an eluent at a flow rate of 1 mL.min^−1^ at room temperature. A multi-detector was used, and the columns were calibrated with standard polystyrene.

### 4.4. Morphological Characterizations

The phase morphologies of the samples were investigated using a JEOL 7600F FEG scanning electron microscope (SEM) with an accelerating voltage of 10 kV. The morphology of all blends was examined after the blending extrusion stage, i.e., before injection molding. The samples were frozen in liquid nitrogen and fractured, after which they were dissolved by THF to remove the PLA for the direct observation (with the exception of pure PLA), and finally coated with gold using an SPI sputter coater, for enhanced conductivity.

The number-averaged radius ($R_{n}$) and the volume-averaged radius ($R_{v}$) of the domains were respectively calculated using the image processing software ImageJ via the following relations:(2)$$Rn=\frac{\sum_{i}n_{i}R_{i}}{\sum_{i}n_{i}}$$
(3)$$Rv=\frac{\sum_{i}n_{i}R_{i}^{4}}{\sum_{i}n_{i}R_{i}^{3}}$$
where $n_{i}$ is the number of the dispersed domains with radius $R_{i}$ obtained from the SEM images. The total number of analyzed domains was at least 100 particles.

### 4.5. Dynamic Mechanical Analysis (DMA)

The thermo-mechanical behavior of the polymers was studied on compression-molded rectangular samples (12.75 mm × 6.25 mm × 0.24 mm) using a dynamic mechanical analyzer (DMA, Q800, TA Instruments) in tensile mode. DMA thermograms were recorded at a constant frequency of 1 Hz, a strain amplitude of 0.01%, at a temperature scan from −40 to 130 °C, at a heating rate of 2 °C/min, and under a nitrogen atmosphere. The temperature at which the loss modulus (E″), named T_α_, showed a peak in DMA thermograms determined and represented the glass transition temperature.

### 4.6. Small Amplitude Oscillatory Shear (SAOS) Measurements

The results of the linear viscoelasticity of the PLA/PHBV blends were obtained from small-amplitude oscillatory shear (SAOS) flows using a stress-controlled rheometer (Discovery Hybrid Rheometer, DHR-2, TA Instruments) with a plate–plate configuration (Φ = 25 mm) at temperatures of 175 °C, 185 °C, and 195 °C. Disks were placed between plates with a gap inferior to 1.5 mm, and left for approximately 3 min at the measured temperature, to melt/relax the polymer before the tests. To measure the thermal stability of the neat polymers, dynamic time sweep tests were first conducted for neat PLA and PHBV at a strain of 5% and frequency of 1 rad/s for 20 min. Then, measurements were performed under a fixed strain amplitude of 2% in the linear viscoelastic regime, from the high angular frequency (ω) of 628 down to low 0.314 rad/s, with five points per decade. Indeed, the rheological measurements were carried out from high to low frequencies, which minimized the residence time and decreased the effect of the thermal degradation of PHBV.

### 4.7. Interfacial Tension Analysis

Palierne’s emulsion model, based on the free energy of the interface, was used to evaluate the parameters with regard to the interfacial tension between the PHBV matrix and the respective dispersed PLA phase from the SAOS data [24,25]. This model can be simplified according to Equations (4) and (5), by assuming that the droplet size distribution is narrow (Rv/Rn≤2) and that the interfacial tension does not depend on the shear or variations in the interfacial area:(4)$$G_{b}^{*}\left(\omega \right)=G_{m}^{*}\left(\omega \right)\frac{1+3\emptyset H^{*}\left(\omega \right)}{1-2\emptyset H^{*}\left(\omega \right)}\;$$
(5)$$\mathrm{With}\;H^{*}\left(\omega \right)=\frac{4\left(\frac{\gamma }{R_{v}}\right)\left[2G_{m}^{*}\left(\omega \right)+5G_{d}^{*}\left(\omega \right)\right]+\left[G_{d}^{*}\left(\omega \right)-G_{m}^{*}\left(\omega \right)\right]\left[16G_{m}^{*}\left(\omega \right)+19G_{d}^{*}\left(\omega \right)\right]}{40\left(\frac{\gamma }{R_{v}}\right)\left[G_{m}^{*}\left(\omega \right)+G_{d}^{*}\left(\omega \right)\right]+\left[2G_{d}^{*}\left(\omega \right)+3G_{m}^{*}\left(\omega \right)\right]\left[16G_{m}^{*}\left(\omega \right)+19G_{d}^{*}\left(\omega \right)\right]}\;$$

Here, ϕ, ω, and γ denote the volume fraction of the droplets with a volume-average radius $R_{v}$, the angular frequency, and the interfacial tension, respectively. $G_{b}^{*}\left(\omega \right)$, $G_{m}^{*}\left(\omega \right)$ and $G_{d}^{*}\left(\omega \right)$ are the norms of the complex modulus of the blend, matrix, and the dispersed phase, respectively.

For sake of comparison, the geometric-mean [26] Equation (6) was used to estimate the interfacial tension between the polymeric matrix and the dispersed phase:(6)$$\gamma_{xy}=\gamma_{x}+\gamma_{y}-2\left[{\left(\gamma_{x}^{d}\gamma_{y}^{d}\right)}^{\frac{1}{2}}+{\left(\gamma_{x}^{p}\gamma_{y}^{p}\right)}^{\frac{1}{2}}\right]\;$$

Here, $\gamma_{x}$ is the surface tension of material x, $\gamma_{x}^{d}$, and $\gamma_{y}^{d}$ are the dispersive and the polar components of the materials, respectively, and $\gamma_{xy}$ is the interfacial tension between phases x and y.

### 4.8. Dielectric Relaxation Spectroscopy (DRS)

Dielectric relaxation measurements were conducted on a dielectric thermal analyzer (LCR Meter, Agilent E4980A, TA Instruments) equipped with an environmental test chamber (ETC, TA Instruments) for temperature control. Dielectric responses were recorded in isothermal condition in the frequency range from 20 Hz to 2 MHz under a constant voltage of 1 V at temperatures steps from 185 °C to −80 °C upon heating, with an increase of 5 °C.

## 5. Results

### 5.1. Measurement of Thermal Stabilities

It is well known that PHBV has thermal degradation, even if at processing temperature. Figure 2 and Figure 3 show the curves of DSC and TGA for neat PLA and PHBV, and the main data are summarized in Table 2. We can see that the pure PHBV started to degrade at around 244 °C, while degradation started at 301 °C for pure PLA. PHBV thus had a lower initial thermal degradation temperature compared to PLA, owing to its lower initial decomposition temperature. It is worth mentioning that the melt temperature of PHBV is higher by almost 20 degrees than PLA. The above results indicate that PHBV had the narrower processing window. Therefore, we first determined the effect of processing conditions and parameters using the measurements of thermal stabilities of the neat polymers, before processing and measurements.

Figure 4 presents the curves of the rheological dynamic time sweep measurements and further analyze the loss in viscosity with time at various processing temperatures from a rheological point of view, and the main data are summarized in Table 3. For these two biodegradable polymers, it comes as no surprise that the lower the temperature, the lower the degree of viscosity loss. At the same time, PLA decreased gradually at different temperatures, whereas PHBV went down sharply, even at lower temperature. For example, after 400 s of thermal treatment, the PLA had around a 20% viscosity loss at the three different temperature, whereas PHBV had 40% at 175 °C, 70% at 185 °C, and 90% at 195 °C. The above measurements fully demonstrate the asymmetry between PLA and PHBV in terms of their thermal instabilities. Based on the abovementioned results and various references [14], we selected the lowest possible processing conditions, i.e., at 175 °C for 3 min at a rotor speed of 100 rpm.

The feasibility of these conditions was further proven by measurement of the molecular weight via size exclusion chromatography (SEC). The results shown in Figure 5 and Table 4 indicated that after twin screw extrusion (E), there was a small loss of molecule weight of 10% for both pure PLA and PHBV when the extrusion temperature was 175 °C, which was deemed an acceptable degree of degradation. However, after both twin screw extrusion (E) and rheological measurements (R), the molecular weight loss of PHBV reached 35.8%, whereas for PLA it was only 19.5%. Therefore, to minimize the residence time as much as possible, and decrease the effect of the thermal degradation of PHBV, all of the rheological measurements were conducted from high down to low frequencies at 175 °C.

### 5.2. Morphology for PLA/PHBV Blends

Figure 6 shows the morphological structures and phase inversion, obtained by SEM, of neat PLA, PHBV, and their blends at various compositions. The pure PLA exhibited the typical brittle fracture of an amorphous polymer, with a smooth and uniform surface. PHBV, on the other hand, displayed an irregular fracture surface, with some holes and cracks, either due to its crystalline structure or due to the thermal-mechanical treatments. Finally, all the blends displayed their own phases, as well as a phenomenon of biphasic separation, proving that the PLA/PHBV pair was not miscible in the solid state.

The compositional dependence was mainly revealed by the specific process of phase inversion of the blends. The blends could be divided into PLA-rich blends, intermediate compositions, and PHBV-rich blends: the PHBV-rich blends displayed a nodular structure, and the “droplet” particle radius became larger when increasing the content of the dispersed PLA phase according to Table 5. For example, the average radius of PLA droplets in the PHBV matrix went from approximately 167 nm (PLA/PHBV = 10/90) to 227 nm (PLA/PHBV = 30/70) [14]. Then, when the PLA content was raised, the droplets in the intermediate compositions (i.e., blends containing 40% and 50% PLA) started to coalesce, and a continuous morphological structure, as well as the point of phase inversion, finally appeared for the PLA/PHBV = 60/40 blend. The PLA-rich blends presented complex and irregular structures, such as ‘S’-type or long-fiber structures after droplets coalescence, as well as a transition state from a nodular to a co-continuous structure when the PHBV contents increased to 30% wt. Finally, when the PHBV content reached 40%, a co-continuous structure also appeared [15].

The mechanisms of the phasic morphology and phase inversion can be explained by the critical capillary number ($Ca_{c}$) calculated via the following Relation (7): [27]
(7)$$lgCa_{c}=-0.506-0.09951lgM_{\eta *}0.124{\left(lgM_{\eta *}\right)}^{2}-\frac{0.115}{lgM_{\eta *}-lg4.08}\;M_{\eta *}=\frac{\eta_{d}}{\eta_{m}}$$

Subsequently, the capillary number (Ca) could be determined by Equation (8):(8)$$Ca=\frac{\eta_{m}\dot{\gamma }R}{\Gamma }$$

Here, $M_{\eta *}$ is the viscosity ratio, $\eta_{d}$ is the viscosity of the dispersed phase, and $\eta_{m}$ is the viscosity of the matrix phase, obtained from the next SAOS rheological results.

The specific morphological structures could be predicted using the values of the viscosity ratio and a comparison between Ca and $Ca_{c}$. When $M_{\eta *}=0.16\ll 1$, and $Ca_{c}$ occurred at a frequency around 10 rad/s, the dispersed PHBV phase was able to coalesce and display the morphology of an ‘S’-type structure or long fibers in the PLA matrix. For PHBV-rich blends, when PLA (less than 30%) was the dispersed phase, $Ca\geq Ca_{c}$, and m=6.1>4, no breakup of the PLA droplets occurred. That is why the dispersed PLA phase displayed a droplet structure. Finally, a co-continuous morphology appeared in blends containing around 40% PHBV, indicating that the point of phase inversion took place in blends with less than 50% PHBV. These results can be explained by the viscosity ratio model, in which the component ratio is proportional to the viscosity ratio [28].

It is worth noting that the blends containing less than 40%wt PHBV showed the complete “nodular” structure reported by Thibaut et al. [14]. In our case, we obtained different results with the same blends. A possible reason for this could be that the extrusion grade PLA (PLA2003D) that we used had a higher viscosity compared to the injection grade (3052D) of their polymer. In addition, the processing conditions, including different temperatures and viscous dissipation owing to high shear rates (rotor speed), affected whether a blend formed coalescence or kept a “droplet” structure in the melt state, under otherwise identical conditions for PLA [29].

### 5.3. Investigation of PLA/PHBV Blend Miscibility

Measuring the glass transition temperature, T_g_, is an effective way to evaluate the miscibility of biphasic blends systems. The T_α_ values of PLA/PHBV blends with varying compositions were obtained from the maximum peaks of E″ in DMA curves, as well as the T_g_ points in DSC curves, shown by the circles and squares, respectively, in Figure 7. We can see that the tendency of the curves measured by DSC quantitatively followed those obtained by DMA, although the values of the two T_α_ values from DMA were higher than those found with DSC [30].

As can be seen, over the whole range of compositions, the blends presented two experimental T_g_ (T_α_) values, confirming that PLA and PHBV were not miscible in the solid state, as has been reported [31]. Based on this, it was found that when the PHBV content was increased, the T_g_ of PLA slightly decreased over the whole range of compositions, while that of the PHBV increased in the intermediate compositions and then reached a plateau in the intermediate compositions and PHBV-rich blends. At the same time, it was clear that the curves of T_g_s of the two phases moved closer for the intermediate compositions and PHBV-rich blends, and the difference in T_g_ between PLA and PHBV also became smaller in these blends, as shown by Figure 8. The above results indicate that, to a certain extent, a relatively better compatibility appeared at intermediate compositions and PHBV-rich blends compared with PLA-rich blends. This is caused by molecular interactions between interphases, and their partial miscibility or compatibility that was unveiled in the solid state over the whole range of compositions.

The specific reasons for the abovementioned phenomena, named in situ self-compatibilization or plasticization mechanisms, were as follows: first, when increasing the PHBV contents, for the PLA phase in blends, adding PHBV (lower T_g_) made it possible for the plasticizer and compatibilizer to plasticize the PLA and increase its mobility, thanks to the small and degradable PHBV molecules that were miscible with PLA [19]. At the same time, the interphase zone between PLA and PHBV also grew, and this was where the entanglements between the two polymer segments took place. The segment motion of the amorphous PLA was facilitated by the soft amorphous PHBV, which could lead to a decreased T_g_ for PLA.

Whereas, for PHBV phase in the blends, this could be explained by the changes of degree of crystallization of PHBV. As can be clearly seen from Table 6, the degree of crystallization of PHBV also increased sharply in PHBV-rich blends and reached a plateau in intermediate compositions and PHBV-rich blends, showing the same tendencies as the curves of T_g_. Therefore, for the blends containing more than 30% PHBV (intermediate compositions and PHBV-rich blends), the degree of crystallization almost reached the maximum and constant values, and the amorphous PHBV molecular could be confined by the surrounding crystalline phase and PLA amorphous phase, all of which led to constant T_g_ values of PHBV. On the contrary, for PLA-rich blends, the PHBV molecules displayed a more amorphous state in these blends and less interphase between PLA and PHBV; therefore, the amorphous PHBV had greater mobility in the segment without a confined state.

In conclusion regarding the miscibility between PHBV and PLA, despite the two polymers not being fully miscible in the solid state over the whole range of compositions, proven by the two T_g_s found experimentally and the biphasic separation, their partial miscibility or compatibility were unveiled by the changes of the two T_g_ values, i.e., the degree of miscibility in almost symmetrical intermediate blends and in the asymmetric PHBV-rich blends being higher than in the asymmetric PLA-rich blends. The above results could be explained by the small differences in solubility parameter values and the fact that a good miscibility was thermodynamically expected, as well as their in situ self-compatibilization or plasticization mechanisms. It could be depicted as degraded in PHBV and the small molecules played an important role as a plasticizer and compatibilizer in improvement of the their compatibility.

### 5.4. Small-Amplitude Oscillatory Shear (SAOS) Rheology

#### 5.4.1. Linear Viscoelastic Properties

The SAOS rheology was first measured in the linear viscoelastic (LVE) regime for PLA, PHBV, and their blends. The fundamental rheological parameters of neat PLA and PHBV are shown in Table 7, indicating that both PLA and PHBV were linear aliphatic polyesters. However, compared with PLA, PHBV had a relatively higher chain regularity, which could be proven by its higher entanglement molecular weight ($M_{e}$), which was defined as the average molecular weight between neighboring entanglement points. Therefore, the chain structures of PLA and PHBV were described as jagged and straight regular structures, respectively, as shown in Figure 9.

Figure 10 displays the evolution of the complex viscosity $\eta^{*}$ (a), storage modulus $G^{\prime }$ (b), loss modulus $G^{\prime \prime }$ (c), and $M_{\eta *}$ viscosity ratio (d) as a function of frequency of the neat PLA, PHBV, and their blends. On the whole, the viscosity curves of all samples almost demonstrated a plateau in the lower and middle frequency regions and a shear-frequency behavior at higher frequencies. Their corresponding $G^{\prime }$ and $G^{\prime \prime }$ values showed an increase with an increase of the angular frequency. All of which are the typical characteristics of linear polymers, as we mentioned before.

First, the neat PHBV and PLA had a huge difference in their complex viscosities at whole frequencies, indicating the asymmetry of viscosities of the PLA/PHBV pair, due to a different chain regularity. For example, at 10 rad/s, PLA presented a viscosity of 4763 Pa·s, while the corresponding value for PHBV was 638 Pa·s. At the same time, they had relatively higher values of viscosity ratio (more than 3) between PLA and PHBV over the whole frequency range, which is further shown in Figure 10d. The above results further indicated the highly mismatched viscoelastic properties between PLA and PHBV, despite $M_{\eta *}$ being decreased when increasing the temperature to 195 °C. Meanwhile, pure PLA exhibited a terminal behavior, with a plateau in the viscosity curves, and a slope of 1 and 2 on the log–log plot of $G^{\prime \prime }$ and $G^{\prime }$ vs. ω, respectively. Pure PHBV, on the other hand, did not follow the same law, because of its heavy degradation.

Next, there was also a difference in terminal zone between the neat polymers and their blends. They also presented Newtonian plateaus at low frequencies, which depended on their compositions, and based on these varying terminal behaviors, all the samples, besides pure PLA, could be divided into three types for the following discussions, i.e., PLA-rich blends, intermediate compositions, and PHBV-rich blends (including pure PHBV).

For the PLA-rich blends, an upward shift of the complex viscosities and a shoulder of $G^{\prime }$ were observed at lower frequencies, which is consistent with previous reports in various immiscible blends [32]. This phenomenon reflects the shape–relaxation response of the dispersed droplets The surface or interface of dispersed droplets in the continuous matrix between PLA and PHBV plays an important role in the viscoelastic response of the blends and leads to an additional elastic response. This phenomenon will be further reflected on in the following discussions on the Han plot and weighted relaxation spectra.

The intermediate compositions, containing intermediate and closed proportions of PLA and PHBV (PLA/PHBV = 40/60, 50/50, and 60/40), almost displayed a plateau and Newtonian terminal behavior, although a shoulder in $G^{\prime }$ can be seen for the frequency. This result could be explained by the effects of both heavy degradation and the “additional” elastic response due to the thermal instability of PHBV as its content was further increased.

For the PHBV-rich blends, including the pure PHBV, a downward shift of the complex viscosity was observed, indicating that the terminal behavior was governed by the significant thermal degradation of PHBV. Although the measurements of the frequency sweeps went from high to low values, the longer residence time at lower frequencies exceeded the thermal degradation. When the temperature was increased to 195 °C, degradation led to a sharp drop at low frequencies.

Generally, it is known that the viscosity of miscible systems should obey the mixing law, as in the additivity rule described in Equation (9):(9)$$log\left(\eta_{0,blend}\right)=\underset{i}{\sum }\emptyset_{i}log\eta_{0,f}\;$$
where $\eta_{0,f}$ and $\emptyset_{i}$ are the zero-shear viscosity and the volume fractions of component (i) in the mixture.

Figure 11a depicts the zero-shear viscosity ($\eta_{0}$) estimated by the Carreau-Yasuda model [33] for pure polymers and their blends at different temperatures. Apparently, the mixing law was not obeyed, especially at lower frequencies, according to the higher degradation of PHBV dispersed phase. Positive deviations for $\eta_{0}$ versus ϕ with respect to mixing rules were observed at 175 °C, giving rise to an upward convex behavior. More interestingly, the viscosities of the PLA-rich blends were higher than the theoretical ones and even led to a maximum at ϕ PLA = 80%. In addition, as the temperature increased to 195 °C, a downward convex behavior of $\eta_{0}$ appeared for the PHBV-rich blends, due to the significant degradation of PHBV.

At the same time, it is worth mentioning that all the studied blends displayed intermediate rheological behaviors between neat PLA and PHBV. The viscosities of the PLA/PHBV blends at 1 rad/s represented a good correlation between the experimental and theoretical values, especially at lower temperatures, as shown in Figure 11b, although the PLA-rich blends and intermediate compositions showed a positive viscosity deviation behavior at 175 °C. Moreover, negative deviation caused by thermal degradation was also observed at higher temperatures.

A Han plot [34] was used to evaluate the miscibility or compatibility of biphasic blends systems in the melt state. A linear correlation in the plot of log $G^{\prime }$ versus log $G^{\prime \prime }$ was considered as the miscible blends pair; otherwise, it was described as an immiscible or phase-separated blend. It is clearly shown by the Han plot, as seen in Figure 12, that the curves of neat PLA and PHBV displayed a linear correlation with a similar slope over the whole modulus; whereas, a nonlinear correlation and an upturned shifting behave appeared at a lower modulus for all the blends, especially for PLA-rich blends, although a linear correlation with a similar slope was displayed at higher modulus (higher frequencies) [35].

The above phenomenon could be explained by the PLA/PHBV pair not being a miscible system in the molten state, and their thermos-rheological complexity was displayed at lower frequency, peaking for PLA-rich blends. Nevertheless, a certain degree of compatibility between PLA and PHBV was still shown from a rheological point of view, due to their similar linear chain structure.

#### 5.4.2. Weighted Relaxation Spectra

The weighted relaxation spectra of all the samples were calculated using a non-linear regularization method [36] at 175 °C with the SAOS data, according to Equation (10):(10)$$G*\left(\omega ,\lambda \right)=\overset{+\infty }{\underset{-\infty }{\int }}\frac{H\left(\lambda \right)i\omega \lambda }{\lambda \left(1+i\omega \lambda \right)}dy$$

Here, “ω” is the angular frequency, “λ” is the relaxation time, H(λ) is the continuous relaxation spectrum, and λH(λ) is the weighted relaxation spectrum.
$$G^{\prime }\left(\omega \right)=\overset{+\infty }{\underset{-\infty }{\int }}H\left(\lambda \right)\frac{\omega^{2}\lambda^{2}}{\lambda \left(1+\omega^{2}\lambda^{2}\right)}d\left(\ln\lambda \right)\;G^{\prime \prime }\left(\omega \right)=\overset{+\infty }{\underset{-\infty }{\int }}H\left(\lambda \right)\frac{\omega \lambda }{1+\omega \lambda }d\left(\ln\lambda \right)\;\eta_{0}=\overset{+\infty }{\underset{-\infty }{\int }}\lambda H\left(\lambda \right)d\left(\ln\lambda \right)$$

Figure 13 displays the weighted relaxation spectrum of the pure polymers and their blends. As can be seen, all the samples exhibited one main peak for relaxation: the mean relaxation time of pure PLA was 0.044 s, whereas that of PHBV was lower at 0.018 s. The relaxation times of the blends decreased as the PHBV content was raised, which could be explained by the fact that PHBV presented more short chains, induced by their lower thermal stability, and that its chains had a more regular chemical structural compared to PLA.

In addition, another peak representing interfacial relaxation between the two phases appeared at more than 1 s for the three PLA-rich blends (PLA/PHBV = 90/10, 80/20, and 70/30). This phenomenon has also been observed for immiscible blends such as PLA/PBAT [9], PLA/PA11 [35], and PLA/PBSA [37], and showed that the interface of dispersed phase in the continuous matrix between PLA and PHBV played an important role in the elastic response of the blends. This could be caused by their long fiber structures after droplets of PHBV coalescence, as we mentioned in the above morphological studies.

#### 5.4.3. Interfacial Properties of the PLA/PHBV Blends

Values of the interfacial tensions between the immiscible PHBV/PLA pair with dispersed PLA were obtained from the SAOS results using the Palierne model (Equations (5) and (6)), assuming that the droplet size distribution was narrow (Rv/Rn≤2) and that the interfacial tension was independent of the shear and the variation of interfacial area. Figure 14 shows the best fitting results of the SAOS response of the 20/80 sample using the Palierne model. We can see that the predicted storage modulus ($G^{\prime }$) and loss modulus ($G^{\prime \prime }$), as well as complex modulus ($G^{*}$), values agreed very well with the experimental data.

From these fits, the interfacial tension between the components was calculated as $\gamma_{\mathrm{PHBV}-\mathrm{PLA}}$ = 0.84 mJ/m^2^ for γ/Rv = 4260 N/m^2^, although this value was lower than the 1.92 mJ/m^2^ from the geometric-mean equation, and two kinds of values were summarized in Table 8.

By comparison, the corresponding theoretical value of interfacial tension between PBS [31] and PLA, when PLA was the dispersed phase, was 1.12 mJ/m^2^, that of the PLA/PBSA and PLA/PBAT pairs were 1.25 and 1.57 mJ/m^2^, respectively, with the PLA matrix, [38] indicating that the PLA had a lower interfacial tension and higher adhesion with PHBV as opposed to PBS, and even other degradable polyesters. The relatively good compatibility of the PLA/PHBV pair is also be proven by the above results, from a rheological point of view.

Therefore, we can draw conclusions for the molten state, from their chemical chain structure and rheological properties. First, PLA and PHBV are similar linear polymers, which brings similar rheological characteristic and a certain degree of compatibility between them, especially at higher frequencies. The above results further prove their partial miscibility and compatibility in the molten state, and the relative lower value of interfacial tension from the SAOS results also supports this point. However, the high chain regularity of PHBV led to its lower viscosity and a huge difference of viscosity with PLA, despite PHBV having a higher molecular weight than PLA. Therefore, they also showed highly mismatched viscoelastic properties. At the same time, the thermal rheological complexity appeared at lower frequencies, especially for the PLA-rich blends, which mainly reflected the upward shifting behavior in the storage modulus and the shape or interfacial relaxation peak in the weighted relaxation spectra.

### 5.5. Dielectric Relaxation Spectrum for PLA/PHBV Blends

Figure 15 displays the curves of the storage and loss permittivity in the temperature range of –80 °C to 185 °C at 1 KHz. Two types of dielectric relaxation, named α-relaxation and Maxwell–Wagner–Sillars (MWS) relaxation [39], could be clearly observed; whereas, the peak of β-relaxation was so weak as to be negligible.

PLA’s α-relaxation had a peak maximum at around 75 °C, which was associated with the long-range cooperative chain motions as the temperature passed the glass transition. The α-relaxation of PHBV [40], located at around 20 °C, corresponded to the segmental chain mobility, associated with the glass transition [41,42]. Regarding the blends, the different types of relaxation associated with the glass transition and sub-T_g_ motions were altered, especially for the α-relaxation of PHBV. The blends with 20% PHBV presented one large and clear peak at around 15 °C, and this peak shifted to around 20 °C when the PHBV content was increased 40% higher. This was in agreement with the aforementioned results that the PLA-rich phase had a lower degree of crystallinity and a T_g_ in the miscible part. However, a weak peak was seen in the vicinity of the T_g_ of PLA and seemed to be covered by strong peaks from the MWS relaxation of PHBV.

Finally, in the zone of the MWS relaxation, we could observe strong peaks for the intermediate and PHBV-rich phases, indicating that interface polarization occurred between the amorphous phase of PLA and PHBV and the crystalline phase of PHBV. By contrast, blends with 20% PHBV presented a weak relaxation peak. The reason for this was the difference in the degree of crystallization at varying PHBV contents, as mentioned above.

Electric modulus formalism is advantageous over other techniques, as it suppresses the effects of electrode frequency. Dielectric relaxation spectroscopy (DRS) was employed to obtain further crucial insights into the molecular dynamics of the PLA/PHBV blends in the melt state. Figure 16 shows the frequency dependence of the dielectric loss modulus ($M^{\prime \prime }$) for neat polymers and blends at temperatures from 185 °C to 150 °C. Both PLA and PHBV are known to be dielectric type-B polymers, which means that they present a conductivity relaxation with a characteristic relaxation time defined by $2\pi f_{max}\tau =1$. The conductivity relaxation peak reflects the transition from long-range to short-range mobility of the charge carriers (catalyst, impurities, etc.) along conductivity paths. We can see that all the samples shifted to higher frequencies with increasing temperature (Figure 16). In contrast to the lower frequencies (10^2^–10^3^ Hz) of PLA (Figure 16a), the conductivity relaxation of PHBV emerged at higher frequencies (10^4^ Hz) (Figure 16d), suggesting a higher mobility and a larger quantity of charge carriers in PHBV. It can also be observed that the characteristic relaxation frequency ($f_{max}$) of conductivity relaxation shifted to higher frequencies with increasing PHBV contents and reached a maximum for the intermediate phase. When the PHBV contents were higher than 40%, PHBV became dominant, and a single characteristic relaxation peak could be seen.

Interestingly, for PLA-rich blends (PHBV ≤ 40%), a shoulder-like relaxation peak emerged in the frequency region below $f_{max}$, particularly at lower melt temperatures (above 150 °C). This shoulder moved toward higher frequencies and came closer to the conductivity relaxation peak with increasing temperature, and was finally hardly discernible at temperatures above 185 °C.

As mentioned above, the emerging relaxation observed at lower frequencies was ascribed to the relaxation of charge carriers at interfaces within the bulk of the sample, a phenomenon known as interfacial Maxwell–Wagner–Sillars (MWS) polarization. It is noteworthy that the MWS relaxation observed in the blends under investigation displayed a composition dependence. Clearly proving this trend, Figure 17 shows plots of the normalized $M^{\prime \prime }$ curves of the blends with $\emptyset_{PHBV}\geq 20\%$ for the conductivity relaxation, i.e., $M^{\prime \prime }/{M^{\prime \prime }}_{max}$ versus $f^{\prime \prime }/{f^{\prime \prime }}_{max}$. Notably, the MWS relaxation peak was the most pronounced for blends with PHBV contents between 20% to 40%, and the charge carriers also demonstrated a bimodal relaxation behavior. The charge carriers and conductivity relaxation were dominated by the PHBV phase with a lower T_g_ and a high degree of crystallization in the blends. Regarding the bimodal relaxation peaks, we discovered a higher peak at the higher frequency transfer, when the PHBV contents were in the range of 20% to 30%. This phenomenon was attributed to interface polarization between the amorphous phase of the PLA or PHBV, and the crystalline parts of PHBV; at the same time, we could observe a sharp increase when the PHBV contents were increased to 20%. The above results could be explained in that the crystallization behavior plays an important role in biphasic separation from a molten to solid state.

## 6. Conclusions

The present work was carried out to gain a true and fundamental understanding of the compositional dependence, regarding the rheology, morphology, miscibility, and thermal stability of PLA/PHBV blends obtained by melt mixing. Blend samples were prepared over the entire range of compositions, and, at first, restricted processing conditions were chosen, due to the inherent thermal degradation of PHBV, as proven by rheological dynamic time sweep (DTS) tests and size exclusion chromatography (SEC). Next, the compositional dependence of the blends’ properties, such as the thermo-rheological complexity and miscibility, morphological structures, and thermal stabilities in the asymmetric and nearly symmetric phases, was investigated and could be further explained by the interfacial properties of the blends. The origin of the compositional dependence was in the disparate properties between PLA and PHBV, on the basis of their immiscibility. A high viscosity ratio led to irregular structure morphologies and could further account for the maximum degree of immiscibility and thermo-rheological complexity that appeared in the asymmetric PLA-rich phases. This could also be seen in the crystal–amorphous interfacial MWS polarization, due to the different crystallization behaviors. By contrast, the miscibility and thermo-rheological simplicity were improved in the intermediate compositions and the PHBV-rich phases at lower temperature, since their properties were, to a larger extent, governed by the heavy degradation of PHBV. This study will help gain a better understanding of the interfacial phenomena and in situ nanostructure of PLA/PHBV blends during multilayer co-extrusion.

## Figures and Tables

**Figure 1 polymers-14-05359-f001:**
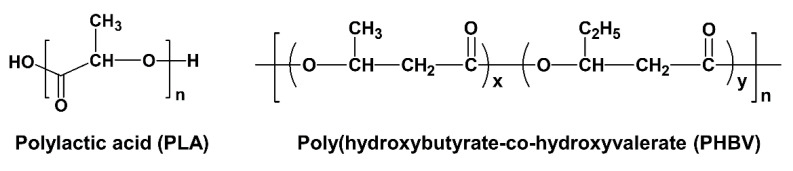
Chemical structure of pure PLA and PHBV.

**Figure 2 polymers-14-05359-f002:**
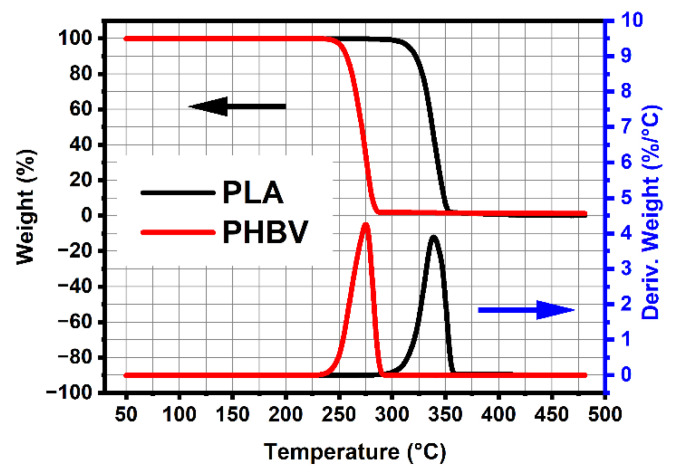
Results from TGA of pure PLA and PHBV.

**Figure 3 polymers-14-05359-f003:**
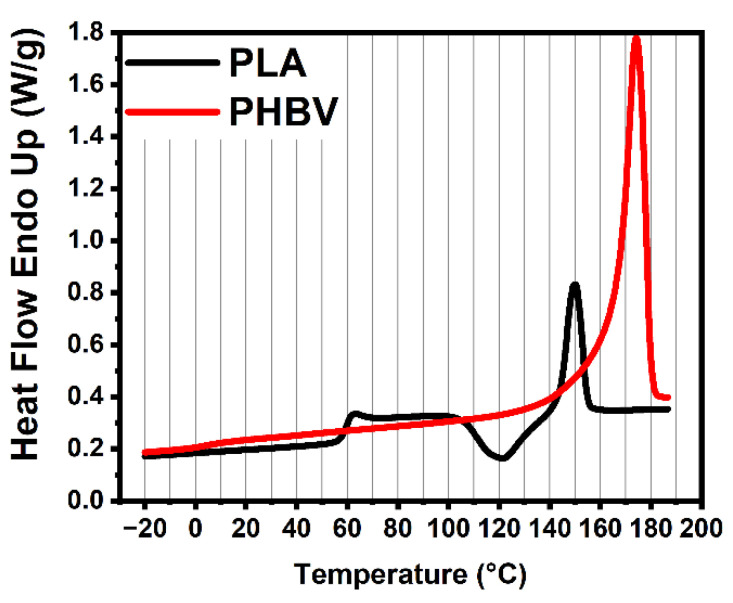
Results from DSC of neat PLA and PHBV.

**Figure 4 polymers-14-05359-f004:**
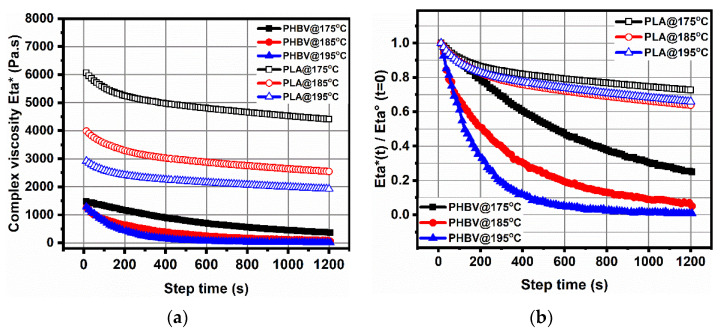
Results from the rheological dynamic time sweep measurements of pure PLA and PHBV at various temperatures. (**a**) Complex viscosity Eta*(t) versus time t; (**b**) complex viscosity divided by zero complex viscosity Eta*(t)/Eta° (t = 0) versus time t.

**Figure 5 polymers-14-05359-f005:**
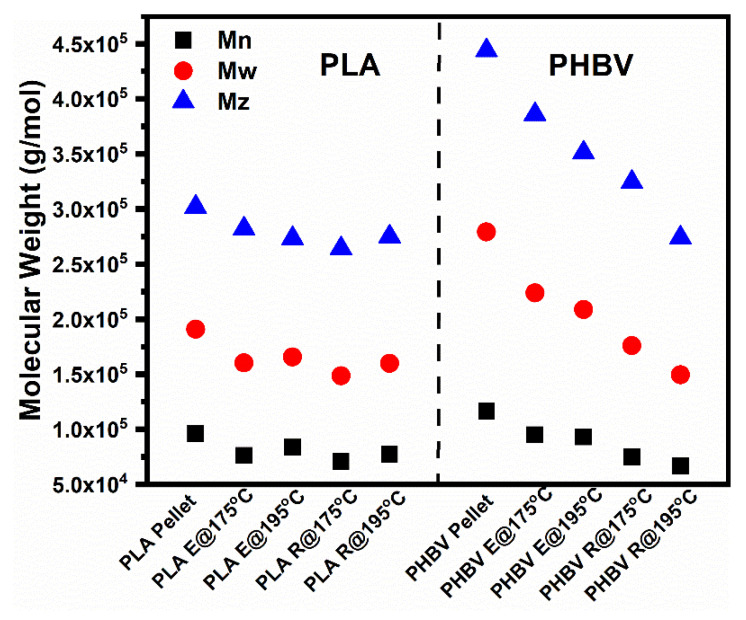
Scatter plots of three molecular weights of PLA, PHBV, and their blends.

**Figure 6 polymers-14-05359-f006:**
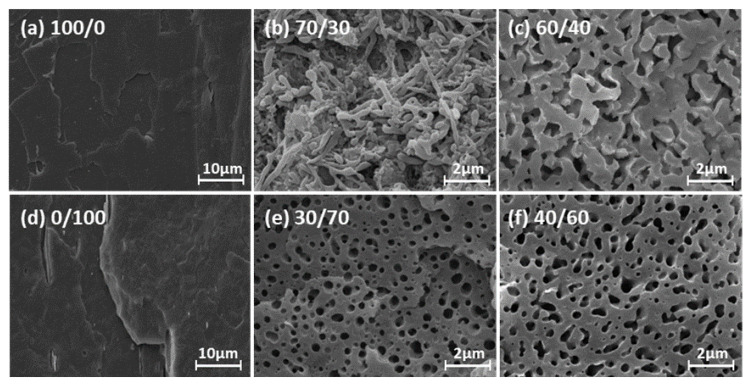
SEM micrographs of fractured surfaces of PLA/PHBV blends at different compositions: 100/0 (**a**), 70/30 (**b**), 60/40(**c**), 0/100 (**d**), 30/70 (**e**), and 40/60 (**f**).

**Figure 7 polymers-14-05359-f007:**
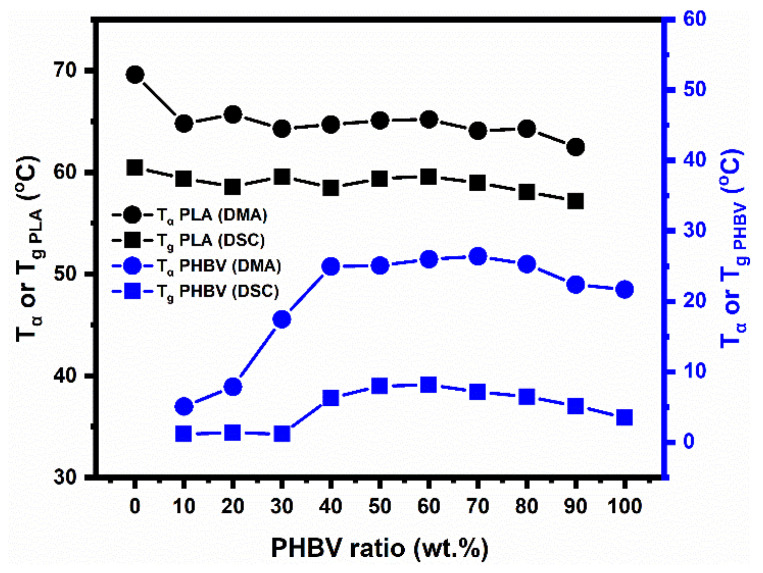
Glass transition temperature of PLA/PHBV blends as determined by DSC (square) and DMA (circles).

**Figure 8 polymers-14-05359-f008:**
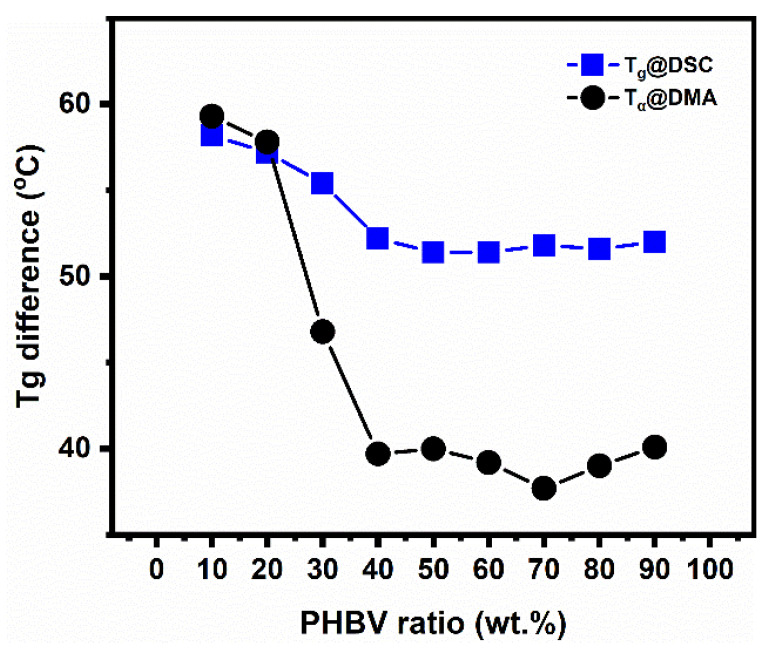
The difference in glass transition temperature values T_g_ between PLA and PHBV.

**Figure 9 polymers-14-05359-f009:**
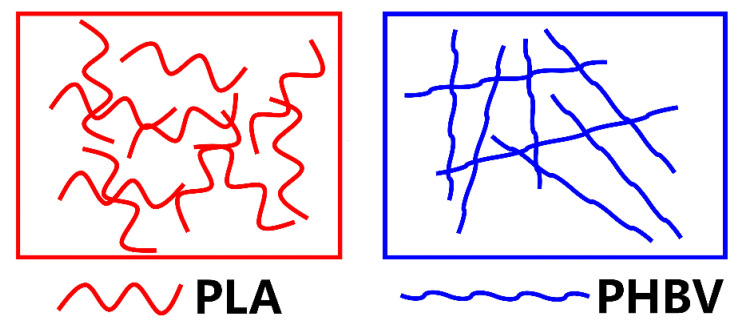
Scheme illustrating of Chain structure of neat PLA and PHBV.

**Figure 10 polymers-14-05359-f010:**
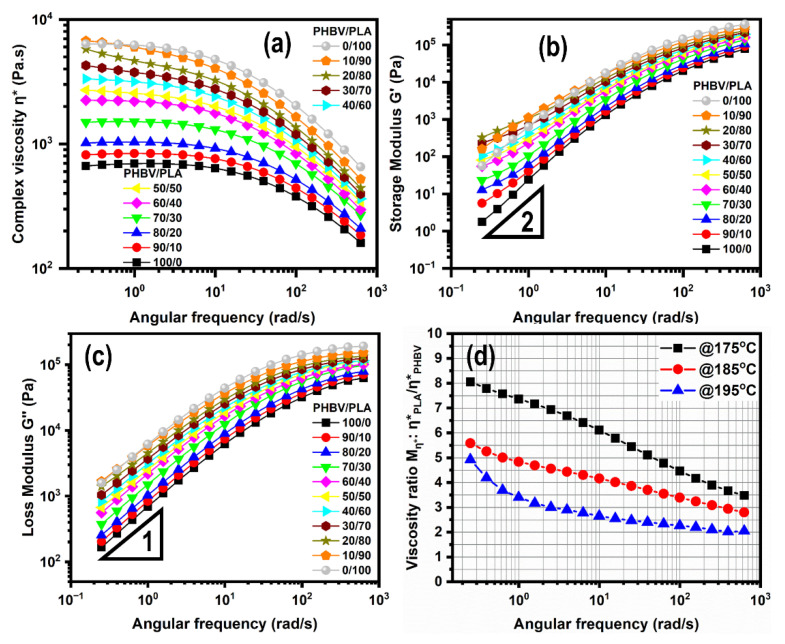
Example of results at 175 °C with (**a**) complex viscosity modulus $\eta^{*}$, (**b**) storage modulus $G^{\prime }$, (**c**) loss modulus $G^{\prime \prime }$, and (**d**) viscosity ratio $M_{\eta *}$ for neat PLA, PHBV, and their blends with various compositions.

**Figure 11 polymers-14-05359-f011:**
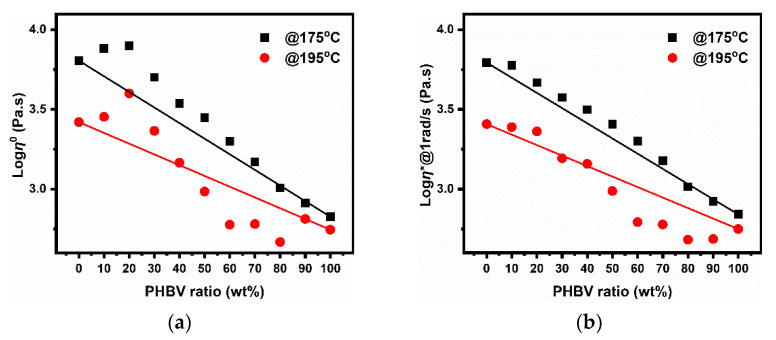
Plots of (**a**) zero-shear viscosity and (**b**) complex viscosity at 1 rad/s versus PHBV composition as a function of in PLA/PHBV blends at 175 °C and 195 °C (the points represent the experimental values, whereas the black and red lines were predicted using the logarithmic additivity rule at these temperatures).

**Figure 12 polymers-14-05359-f012:**
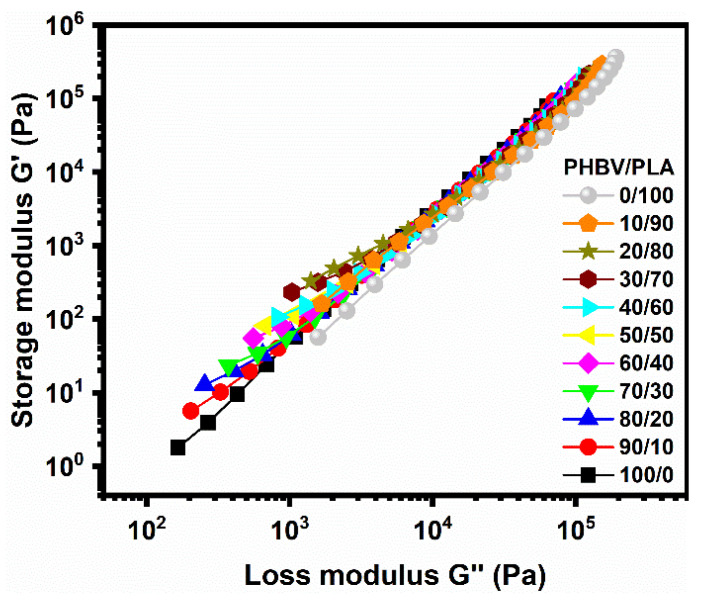
Han plot diagram giving the storage modulus ($G^{\prime }$) versus loss modulus ($G^{\prime \prime }$) for the PLA/PHBV blends at 175 °C.

**Figure 13 polymers-14-05359-f013:**
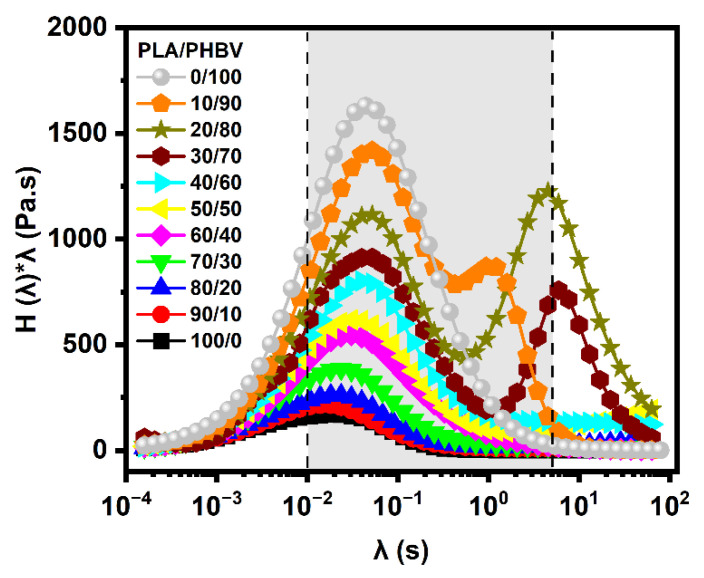
Weighted relaxation spectra for neat PLA, PHBV, and their blends at 175 °C. (Experimental values were located in the grey zone from 0.01 s to 5 s).

**Figure 14 polymers-14-05359-f014:**
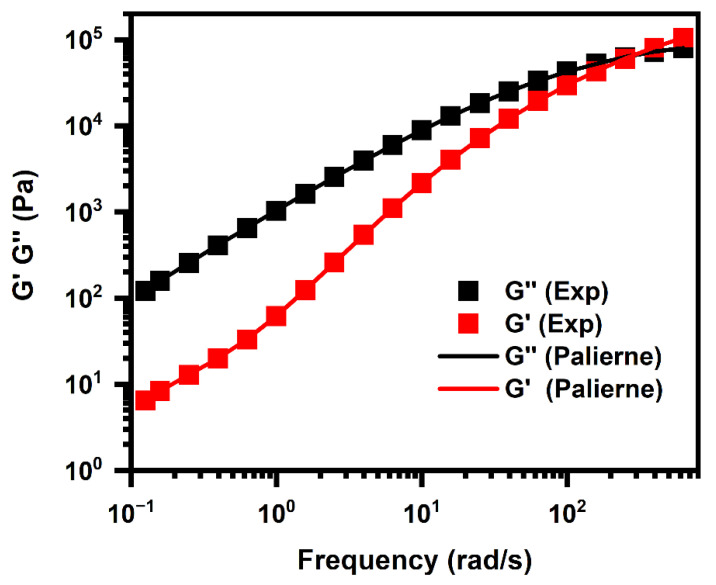
Experimental $G^{\prime }$ and $G^{\prime \prime }$ with the fitted simple Palierne model (solid lines) for PLA/PHBV = 20/80 at 175 °C.

**Figure 15 polymers-14-05359-f015:**
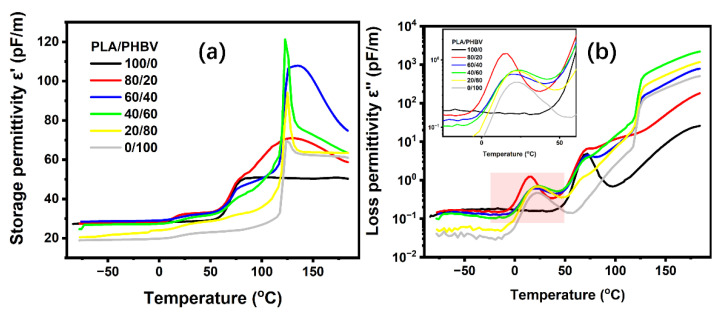
Temperature dependence of the storage permittivity ($\varepsilon^{\prime }$) (**a**) and loss permittivity ($\varepsilon^{\prime \prime }$) (**b**) for PLA, PHBV, and their blends at 1 kHz.

**Figure 16 polymers-14-05359-f016:**
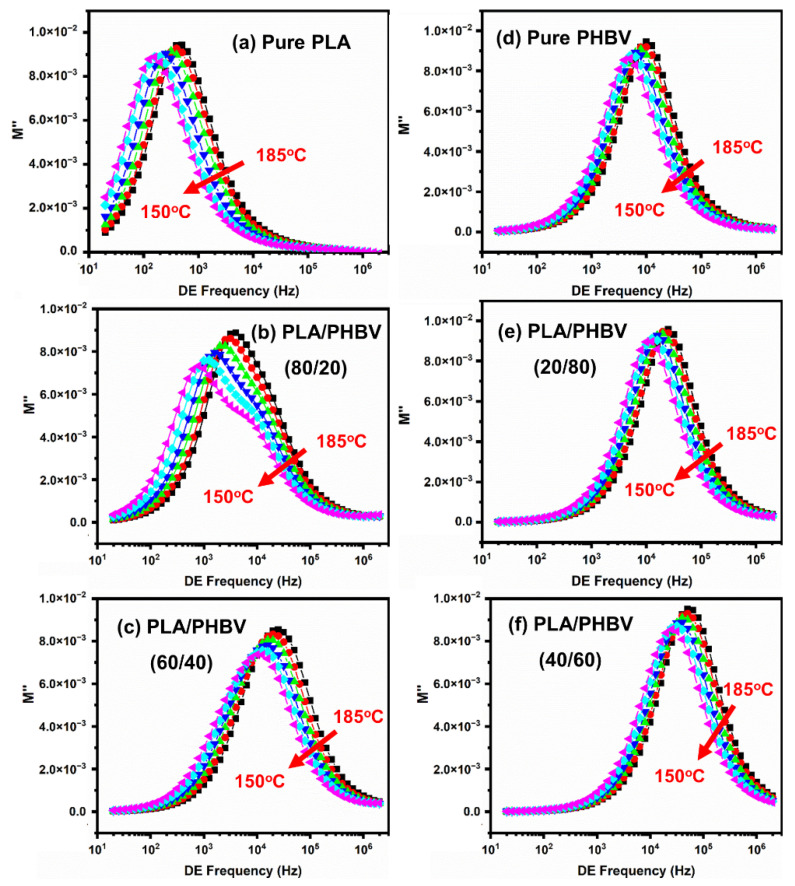
Dielectric loss modulus $M^{\prime \prime }$ as a function of frequency in the temperature range of 185 to 150 °C for (**a**) pure PLA; (**b**) PLA/PHBV (80/20); (**c**) PLA/PHBV (60/40); (**d**) Pure PHBV; (**e**) PLA/PHBV (20/80); (**f**) PLA/PHBV (40/60).

**Figure 17 polymers-14-05359-f017:**
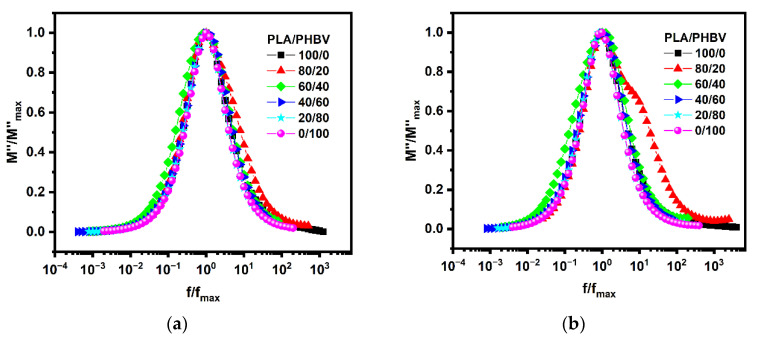
Normalized frequency dependence of $M^{\prime \prime }$ for the conductivity relaxation of PLA/PHBV blends with various compositions at 185 °C (**a**) and 150 °C (**b**).

**Table 1 polymers-14-05359-t001:** Information and properties of PLA and PHBV.

Samples	Supplier	T_g_ (°C)	Tm (°C)	MI (g/10 min (190 °C/2.16 kg))	Density (g/cm^3^)
PLA (2003D)	NatureWorks, Plymouth, MN, USA	56	145–155	5	1.25
PHBV (Enmat Y1000P)	Tianan, Ningbo, China	5	165–175	15	1.25

**Table 2 polymers-14-05359-t002:** Results from DSC and TGA of neat PLA and PHBV.

Samples	DSC		TGA	
	T_g_ (°C)	T_m_ (°C)	T_Onset_ (°C)	T_Peak_ (°C)
PLA	60	150	301	339
PHBV	5	170	244	275

**Table 3 polymers-14-05359-t003:** Results of the rheological dynamic time sweep measurements of pure PLA and PHBV at various temperatures.

Samples	Eta*(t)/Eta^o^ (t = 0) (%)					
	200 s	400 s	600 s	800 s	1000 s	1200 s
PLA@175 °C	87	82	79	77	75	73
PLA@185 °C	83	78	74	71	68	66
PLA@195 °C	82	76	72	69	66	64
PHBV@175 °C	78	60	46	37	30	25
PHBV@185 °C	51	30	19	13	9	5
PHBV@195 °C	33	11	5	3	1	0.8

**Table 4 polymers-14-05359-t004:** Molecular weights and loss rates of pure PLA and PHBV after processing at various temperatures.

Samples	Mn	Loss (%)	Mw	Loss (%)	Mz	Loss (%)	Polydispersity
PHBV pellet	116,619		27,9303		444,098		2.40
PHBV E@175 °C	107,659	7.68	254,686	8.81	414,704	6.62	2.37
PHBV E@195 °C	93,181	20.1	208,811	25.2	351,135	20.9	2.24
PHBV E@175 °C R@175 °C	74,834	35.8	176,031	37.0	324,600	26.9	2.35
PHBV E@175 °C R@195 °C	66,961	42.6	149,584	46.4	274,179	38.3	2.23
PLA pellet	96,278		191,067		301,996		1.98
PLA@175 °C	89,897	6.63	175,243	8.28	282,750	6.37	1.95
PLA@195 °C	83,776	13.0	165,795	13.2	273,597	9.4	1.98
PLA E@175 °C R@175 °C	77,530	19.5	159,903	16.3	275,111	8.9	2.06
PLA E@175 °C R@195 °C	70,813	26.4	148,616	22.2	264,357	12.5	2.10

**Table 5 polymers-14-05359-t005:** Two average radii for blends with different PLA contents.

PLA (%)	*Rv* (μm)	*Rn* (μm)
10	0.179	0.167
20	0.197	0.176
30	0.259	0.227

**Table 6 polymers-14-05359-t006:** Enthalpy and degree of crystallization of PLA and PHBV at different compositions.

Samples (PLA/PHBV)	ΔH_mPLA_ (J·g^−1^)	X_cPLA_ (%)	ΔH_mPHBV_ (J·g^−1^)	X_cPHBV_ (%)
100/0	3.3	3.7		
90/10	2.5	2.8	0.7	4.9
80/20	1.1	1.2	7.7	26.2
70/30	0.5	0.6	22.1	50.5
60/40	4.2	4.7	33.0	56.5
50/50			46.3	63.4
40/60			56.8	64.8
30/70			66.3	64.9
20/80			76.3	65.3
10/90			84.5	64.3
0/100			93.9	64.3

**Table 7 polymers-14-05359-t007:** Fundamental rheological parameters of PLA and PHBV at 175 °C.

Polymers	Mw (kg/mol)	$G_{N}^{0}\;\;(\mathrm{Pa})$	$M_{e}\;(\mathrm{kg}/\mathrm{mol})$
PLA	190	4.7×10^5^	9.5
PHBV	280	2.7×10^5^	13.8

**Table 8 polymers-14-05359-t008:** Interfacial tension of PLA/PHBV using geometric-mean equation and Palierne’s emulsion model.

	γ ^PLA-PHBV^ (mJ/m^2^)
Geometric-mean equation	1.92
Emulsion model of Palierne	0.84

## Data Availability

The data presented in this study are available on request from the corresponding author.

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
