# Peer review of "Study of Morphology, Rheology, and Dynamic Properties toward Unveiling the Partial Miscibility in Poly(lactic acid)—Poly(hydroxybutyrate-co-hydroxyvalerate) Blends"

_polymers, 2022, doi:10.3390/polym14245359_

Round 1

Reviewer 1 Report

It is a very important and sound work, I recommend acceptance for publication after addressing the following minor issues.

If possible, I suggest the authors to add the origin of Eq. (6) by citing a reference.

Table 4 & Figure 5, PHBV pellet and PHBV P, as well as PLA pellet and PLA P, are both used. Could the authors make them identical?

Table 5, please add the definition of Rc summarized in Table 5.

Table 7, The style of Table 7 is not consistent with others, I suggest the authors to make them identical.

Figure 10, please add the legends for the curves shown in Figure 10a. In addition, please add the labels of y-axis in Figure 10b.

Figure 16, I suggest the authors to revise the order of the figures shown in Figure 16, such as d-f instead of d-f-e.

I suggest the authors to cite some recently paper on PLA based materials, such as Coatings 2021, 11, 790

There are some minor grammar errors or typos, please double check the manuscript and correct them. Such as

a) Poly (lactic acid) (Ingeo 2003D) is produced by NatureWorks Co. Ltd., USA was a commercial product. I suggest the authors to delete “is”.

b) Figure 4 and Table 3 presents the curves of the rheological dynamic time sweep measurements, “presents” should be changed to “present”.

Author Response

Response to Reviewers

Dear Reviewers,

We greatly thank you for your assessment of our manuscript entitled “Study of morphology, rheology and dynamic properties toward unveiling the partial miscibility in poly(lactid acid) - poly(hydroxybutyrate-co-hydroxy valerate) blends.” (polymers-2055354).

Sincerely, we thank all reviewers for recommending our manuscript for publication in this journal. We appreciate all the reviewers for their insightful comments and suggestions.

Please find in the revised manuscript, we have taken into consideration the constructive remarks to clarify and make review more up-to-date and clear. An effort was made to answering point-by-point the reviewers’ comments. Besides, the English expressions of the manuscript has been polished and double-checked. All corrections appear in red color in the revised manuscript.

Thank you in advance to consider the manuscript being finally accepted for publication in Polymers Journal

Sincerely,

The Authors.

Please find enclosed the answers to the comments from you as following:

It is a very important and sound work, I recommend acceptance for publication after addressing the following minor issues.

Response:

We appreciate for your positive and constructive comments and suggestions on our manuscript.

If possible, I suggest the authors to add the origin of Eq. (6) by citing a reference.

Response:

Thanks for your kind and useful suggestion. We have found the original paper of equation and supplemented it as a reference.

Table 4 & Figure 5, PHBV pellet and PHBV P, as well as PLA pellet and PLA P, are both used. Could the authors make them identical?

Response:

Thanks for your carefulness, they have been unified into “PLA pellet” and “PHBV pellet”, and revised in Table 4 and Figure 5.

Table 5, please add the definition of Rc summarized in Table 5.

Response:

Thanks for your suggestion. Actually, after our careful checking, we found Rc could not be defined as any precise equations or expressions like Rn and Rv. Besides we thought Rn and Rv can be enough to well describe the average radius of nodular structure. So we have deleted the relevant data and description about Rc.

Table 7, The style of Table 7 is not consistent with others, I suggest the authors to make them identical.

Response:

Thanks for your reminding, we have corrected carefully the font size of first line in Table 7 to be consistent with others.

Figure 10, please add the legends for the curves shown in Figure 10a. In addition, please add the labels of y-axis in Figure 10b.

Response:

Thanks for your suggestion, we have carefully supplemented the legends for the curves in Figure 10a and added the labels of y-axis in Figure 10b according to your suggestions.

Figure 16, I suggest the authors to revise the order of the figures shown in Figure 16, such as d-f instead of d-f-e.

Response:

Very careful reminding, we have changed the order of the figures in both Figure 16 and its explanatory caption.

I suggest the authors to cite some recently paper on PLA based materials, such as Coatings 2021, 11, 790

Response:

Thanks for your suggestion, after reading this article, we thought this excellent and recent study on mechanical properties of PLA can help to support the introduction section of our manuscript. So we have cited it be a reference in our manuscript.

There are some minor grammar errors or typos, please double check the manuscript and correct them. Such as

  1. a) Poly (lactic acid) (Ingeo 2003D) is produced by NatureWorks Co. Ltd., USA was a commercial product. I suggest the authors to delete “is”.
  2. b) Figure 4 and Table 3 presents the curves of the rheological dynamic time sweep measurements, “presents” should be changed to “present”.

Response:

Thanks for your suggestion, we checked carefully the English expression the whole manuscript to make sure they are right and corrected several grammar errors including the above you mentioned.

Reviewer 2 Report

It is a good work, practically publishable. Only a review of the layout of the text is recommended

Author Response

Response to Reviewers

Dear Reviewers,

We greatly thank you for your assessment of our manuscript entitled “Study of morphology, rheology and dynamic properties toward unveiling the partial miscibility in poly(lactid acid) - poly(hydroxybutyrate-co-hydroxy valerate) blends.” (polymers-2055354).

Sincerely, we thank all reviewers for recommending our manuscript for publication in this journal. We appreciate all the reviewers for their insightful comments and suggestions.

Please find in the revised manuscript, we have taken into consideration the constructive remarks to clarify and make review more up-to-date and clear. An effort was made to answering point-by-point the reviewers’ comments. Besides, the English expressions of the manuscript has been polished and double-checked. All corrections appear in red color in the revised manuscript.

Thank you in advance to consider the manuscript being finally accepted for publication in Polymers Journal

Sincerely,

The Authors.

Please find enclosed the answers to the comments from you as following:

It is a good work, practically publishable. Only a review of the layout of the text is recommended

Resonse:

 We appreciate for your decision and comment on our manuscript. We checked carefully the review of the layout of the text, and adjusted the orders between each sections in the whole manuscript to make logically manuscript better.

Reviewer 3 Report

 Lamnawar and coworkers have studied various properties such as  rheology, morphology, miscibility and thermal stabilityof poly(lactid acid) - poly(hydroxybutyrate-co-hydroxy valerate) blend. The work is well planned and presented. Therefore recommend for acceptance with minor revision.

1) In “Introduction section” page 1, line 39,  relatively low price: reviewer is not satisfy this sentence, indeed PLA and other biodegradable polymers are much expensive than polyolefins especially polyethylene and polypropylene. Introduction part need to revise and need to cite recent references (Polymers 2020, 12, 2365, Chem. Commun., 2019,55, 10112-10115, Coordination Chemistry Reviews 414 (2020) 213296).

2) Page 2, line 54: mixing law medium, it would be low medium 

Author Response

Response to Reviewers

Dear Reviewers,

We greatly thank you for your assessment of our manuscript entitled “Study of morphology, rheology and dynamic properties toward unveiling the partial miscibility in poly(lactid acid) - poly(hydroxybutyrate-co-hydroxy valerate) blends.” (polymers-2055354).

Sincerely, we thank all reviewers for recommending our manuscript for publication in this journal. We appreciate all the reviewers for their insightful comments and suggestions.

Please find in the revised manuscript, we have taken into consideration the constructive remarks to clarify and make review more up-to-date and clear. An effort was made to answering point-by-point the reviewers’ comments. Besides, the English expressions of the manuscript has been polished and double-checked. All corrections appear in red color in the revised manuscript.

Thank you in advance to consider the manuscript being finally accepted for publication in Polymers Journal

Sincerely,

The Authors.

Please find enclosed the answers to the comments from you as following:

Lamnawar and coworkers have studied various properties such as  rheology, morphology, miscibility and thermal stabilityof poly(lactid acid) - poly(hydroxybutyrate-co-hydroxy valerate) blend. The work is well planned and presented. Therefore recommend for acceptance with minor revision.

1) In “Introduction section” page 1, line 39,  relatively low price: reviewer is not satisfy this sentence, indeed PLA and other biodegradable polymers are much expensive than polyolefins especially polyethylene and polypropylene. Introduction part need to revise and need to cite recent references (Polymers 2020, 12, 2365, Chem. Commun., 2019,55, 10112-10115, Coordination Chemistry Reviews 414 (2020) 213296).

2) Page 2, line 54: mixing law medium, it would be low medium

Response:

Thanks for your decision as well as suggestions and comments.

  • Indeed, like you said, the price of PLA is much more high than some general plastics such as PP and PE. But to the best of my knowledge, PLA is a widespread used bio-degradable polymer, meanwhile, it has the relatively low price compared with other bio-degradable polymer like PBAT, PBS, PHA and PCL Therefore, I think “relatively low price” written here is basically correct, and it do not need to be revised any more. Besides, I have also read similar expressions from some articles such as Materials Science and Engineering. 2020, 864(1): 012154, and Nanotechnology, 2021, 32(38): 385601.

Recent references are necessary to make our manuscript better. We have read carefully these articles you mentioned, we think these excellent and recent studies on synthesis of PLA can help to support the introduction section of our manuscript. So we have cited them be the references in our manuscript.

  • Thanks for your reminding, we have revised the expression carefully according to your suggestions.
